# India's potential for integrating solar and on- and offshore wind power into its energy system

Tianguang Lu [1,2], Peter Sherman [3], Xinyu Chen[4✉], Shi Chen[5], Xi Lu [5] & Michael McElroy [2,3✉]

This paper considers options for a future Indian power economy in which renewables, wind and solar, could meet 80% of anticipated 2040 power demand supplanting the country's current reliance on coal. Using a cost optimization model, here we show that renewables could provide a source of power cheaper or at least competitive with what could be supplied using fossil-based alternatives. The ancillary advantage would be a significant reduction in India's future power sector related emissions of $CO_2$. Using a model in which prices for wind turbines and solar PV systems are assumed to continue their current decreasing trend, we conclude that an investment in renewables at a level consistent with meeting 80% of projected 2040 power demand could result in a reduction of 85% in emissions of $CO_2$ relative to what might be expected if the power sector were to continue its current coal dominated trajectory.

[1] School of Electrical Engineering, Shandong University, Jinan 250061, China. [2] School of Engineering and Applied Sciences, Harvard University, Cambridge, MA, USA. [3] Department of Earth and Planetary Sciences, Harvard University, Cambridge, MA, USA. [4] School of Electrical and Electronic Engineering, Huazhong University of Science and Technology, Wuhan, China. [5] School of Environment, State Key Joint Laboratory of Environment Simulation and Pollution Control, Tsinghua University, Beijing, China. ✉email: xchen2019@hust.edu.cn; mbm@seas.harvard.edu

With emissions of 2.5 Gt $CO_2$ in 2017, India ranked third globally, trailing only China (9.8 Gt) and the US (5.3 Gt). Coal accounts for the bulk of India's contemporary primary energy supply, 58.1% in 2015[1], and is projected to continue to play an important role indefinitely into the future, 42–50% by 2047[2]. The share of electricity in the overall energy system is predicted to rise from the current level of 16 to 25–29% in 2047 [ref. [2]; indicated also in Supplementary Fig. 1a]. In absolute terms, the demand for electricity is expected to increase by as much as a factor of 4 over this time period.

The capacity for power generation in India amounted to 344 GW in 2018 of which coal accounted for 197 GW (57%), hydro 49.8 GW (14%), wind 34.0 GW (10%), gas 24.9 GW (7%), and solar 21.7 GW (6%) with the balance represented by a combination of biomass 8.8 GW (3%) and nuclear 6.8 GW (2%) [ref. [3]; indicated also in Supplementary Fig. 1b].

The capacity factor (CF) is defined as the fraction of power generated by a particular facility relative to its nameplate potential. Capacity factors for renewable sources are typically much lower than those for coal, gas and nuclear plants given the intermittent nature of the energy sources for the former. Renewables accounted for <7.6% (1.3 PWh) of the total power consumed by India in 2018. The dominant source was supplied by $CO_2$-emitting coal-fired plants. A variety of analyses[4–7] suggests that this trend is likely to continue at least for the immediate future[3]. The locked-in capacity from coal-fired power plants will affect the optimal generation mix as well as the system economics in the future. NITI Aayog (a policy think tank for the Indian government) set a target of 175 GW of renewable capacity for 2022, 160 GW of which would be in the form of either wind or solar[2]. Following these considerations, assessing feasible renewable pathways to decarbonize India's energy sector offers an important and urgent challenge.

This paper considers the possibility of much higher levels of renewables for India in the future. For present purposes, we refer to the combination of wind and solar as renewables. There is a clear need for an integrated view of the potential for a low-carbon future in India. This paper represents for the first time an integrated view of all components of India's electricity system involving wind, solar, hydro, coal, gas, storage, and interregional transmission to meet power demand on hourly basis. An integrated high renewable-planning model is developed here, incorporating both thorough assessment of the potential for renewables accounting at the same time for the practical operational limitations of power systems. Detailed estimates for the physical (cost unconstrained) potentials for wind (onshore and offshore) and solar PV are conducted. Selecting 2040 as the target year, it explores scenarios in which renewables could account for up to 80% of total power demand. The overall objective is to identify the least cost options to satisfy targets for incorporation of specific levels of renewables in the overall power system. Five regional grids are considered and the paper addresses requirements for power for each of these grids on an hourly basis over a typical year. To this end, we allow for modest expansion of the thermal generator fleet as required to compensate for the intrinsic variability of renewable sources, for investments in the interregional transmission grid to facilitate transfer of power from renewable-rich to renewable-poor regions, and for investments in storage systems to enable transfer of power from times of excess to times of deficit.

The primary challenge in planning for power systems under high levels of renewables is to reconcile the conflict between the variability of renewable sources and the intrinsic inflexibility of thermal power systems. This has led to a loss of more than $10 billion[8] for China due to curtailments of wind and solar power. Important contributions have been conducted for national level

pathway studies[9–14]. However, the flexibility issue has been largely simplified, resulting in over optimistic projections of carbon abatement costs and under investments in flexible power generating resources. Here we introduce an integrated renewable energy system planning model designed to co-optimize investments for generation, transmission and storage expansion with detailed treatment of system operations considering not only requirements for balancing supply and demand, but also hourly ramping, reserves, minimal load, and timing involved in start-up and shut down of thermal units. Significant computational challenge is incurred when modeling the full set of flexibility requirements. To accelerate the optimization process, our previously developed Fast Unit Commitment model[15] is applied to reduce the computational complexity for operational simulation.

## Results

**Modeling and simulations.** Incorporating hourly power demand data for five regional grids, high resolution assessments of wind and solar resources, and information for all existing and planned thermal units in India, the analysis indicates that investments in wind and solar could provide a cost competitive alternative to what could otherwise develop as a coal dominated future for India's power system while contributing at the same time to a reduction of as much as 80% in emissions of $CO_2$.

The objective of this study is to identify the least cost options to accommodate a specified fraction of renewable energy in the overall power system for India in 2040. The analysis considers the need to meet the demand for power on an hourly basis over the course of a year for five selected regions of the country: north, west, south, east, and northeast. The data employed for this purpose were developed based on patterns of consumption observed in 2016[16], scaled to allow for the higher levels of demand projected for 2040. All of the results displayed in the following sections were obtained using what we define as the standard model with distinguishing parameters summarized in Supplementary Table 5. We recognize that there are other options that might contribute to a lower carbon future for India including increased reliance on hydro, nuclear, and potentially biomass, in addition to targeted investments to improve energy efficiency. We choose to focus here on wind and solar recognizing the emphasis that has been placed on these resources most recently by the Indian government. Current policy calls for 175 GW of renewable energy by 2022, 160 GW of which would be supplied in the form of either wind or solar. Reports indicate that this focus on wind and solar is likely to continue and indeed to expand beyond this initial target date[3].

Our estimate for the potential source of power from wind in India assumes deployment of a fleet of 2.5 MW Goldwind turbines onshore, with larger, 8.0 MW Vestas, turbines designated for placement offshore. Properties of the turbines selected, including relevant power curves, are summarized in Supplementary Table 1 and Supplementary Fig. 2. The yield of power from individual turbines was calculated on an hourly basis over the course of a year using data from the MERRA-2 reanalysis product available from NASA's Goddard Earth Sciences Data and Information Services Center[17]. This source has a spatial resolution of 0.5° longitude by 0.67° latitude. The approach used to calculate wind speeds at elevations appropriate for the rotor blades and to estimate relevant annual mean CFs is described in the "Methods" section.

To evaluate the potential yield of power from utility scale PV, we followed the approach outlined by Chen et al.[18]. The solar data used in the present study were taken from the NASA GEOS-5 FP database which reports hourly temperatures and incident solar radiation with a spatial resolution of 0.25° latitude by 0.31°

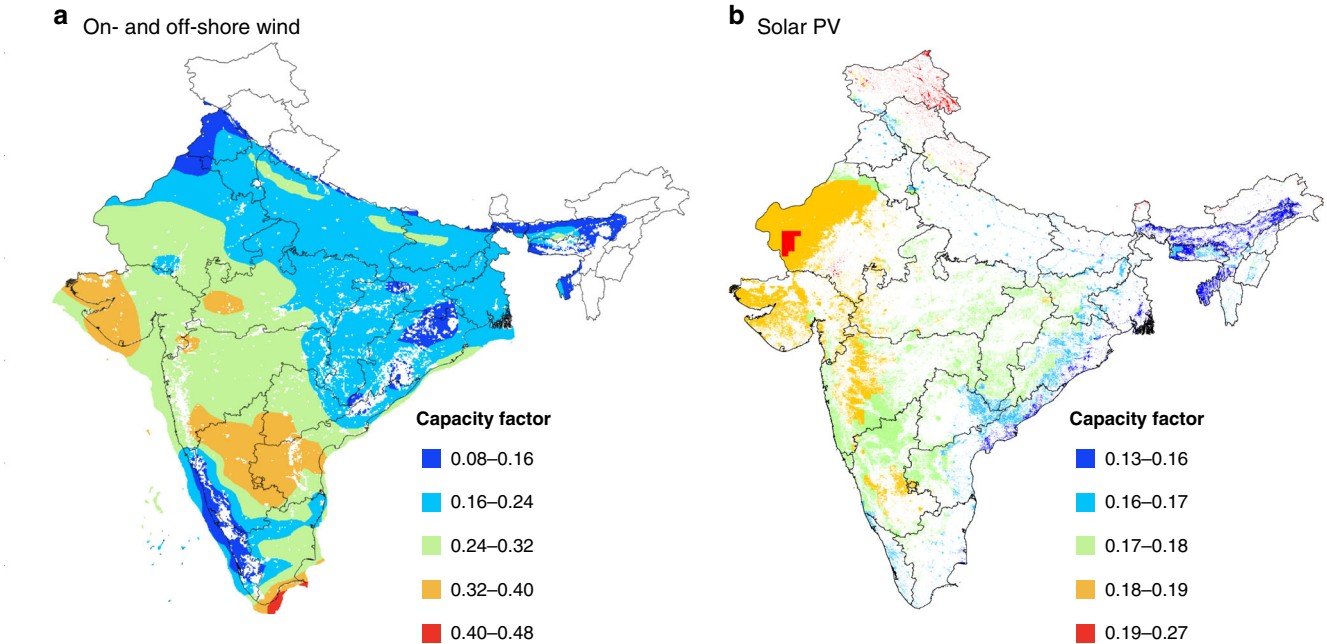

**Fig. 1 Renewable capacity factors.** Spatial distributions of mean annual capacity factors for **a** on- and offshore wind and **b** solar PV constructed based on meteorological output from the MERRA-2 reanalysis product. Regions indicated in white were excluded for the reasons discussed in the text. This figure was constructed in MATLAB version 2017b and edited in Adobe InDesign 2020.

longitude. The spatial variation of factors that can impact relevant CFs was modeled consistently including panel tilt, packing density, sun shading, and temperature. Further details of the computational approach are presented in the "Methods" section.

The potential sources of wind and solar PV, expressed in terms of annual mean capacity factors, calculated subject to the constraints noted above, are displayed in Fig. 1a, b, respectively. As indicated, favorable conditions for wind and solar are confined generally to the west, south, and interior. Less favorable conditions are identified with the northeast. Opportunities for offshore wind are relatively limited, restricted primarily by the 60 m constraint imposed for maximum permissible water depths.

Overall, the physical annual potentials for offshore wind, onshore wind, and solar PV are estimated at 1546, 22,200, and 20,900 TWh, respectively. In projecting future demand for power, we assumed a growth rate of 6.5% per year leading to requirements for a source of 3800 TWh in 2040. The combination of available onshore wind and solar PV, according to the current analysis, should be more than sufficient to account for any conceivable long-term demand for power in India, with a margin of safety that could allow for displacement of fossil sources of energy in other segments of the Indian economy with benefits in terms of further reductions in emissions of $CO_2$.

The optimization model, discussed in the "Methods" section below and further in the Supplementary Information, is designed to identify the most cost-effective options for generation of future power. In the case of costs for thermal generators, this includes accounting for the expense associated with capital investments, costs for operation and maintenance, costs for fuel, and the costs associated with requirements for systems to start up and shut down and to ramp up and ramp down in response to changing demands for power. Properties of the coal and gas-fired plants considered here are summarized in Supplementary Table 4, using data adapted from the World Energy Environment 2019[19]. Geographical distributions and combined capacities of the existing thermal and nuclear power plants in India are presented in Supplementary Fig. 3.

In what follows, we choose to emphasize results from the standard model. Individual models are distinguished by the specific choice of costs for on and offshore wind and for PV, costs for fuels consumed by coal and gas systems, costs for expansion of the transmission network, and costs for storage. The sensitivity of results to the particular choice of input data is discussed in the Supplementary Information and Supplementary Data 1. Data associated with the different simulations are summarized in Supplementary Table 5.

**Regional results**. Conclusions with respect to optimal investments for regional capacities in coal, gas, solar, wind, storage, and for the transmission network corresponding to the 80% renewables scenario are summarized in Fig. 2. The contributions of individual sources to the 2040 power supply consistent with the model, accounting additionally for interregional transfer, are summarized in Fig. 3.

The analysis indicates a significant optimal concentration of investments for wind in the south (521 GW) and west (456 GW), with a comparable concentration for solar in the east (339 GW) and north (227 GW) reflecting the relative endowment of resources for these different regions (Fig. 1). Coal is identified as a critical source for baseload power for four of the five regions considered: west (98 GW), south (44 GW), east (43 GW), and north (41 GW). Storage plays a significant role in the northeast, accounting for as much as 20% of the overall capacity assigned to this region. The absolute magnitude of the commitment (8 GW) is relatively small, however, compared with the related commitment for the north (26 GW). The results indicate that the optimal investment for the Northeast region is solar plus storage, since this region is relatively isolated. Expansion of the transmission network is notably important in facilitating transfer of power between the east and south (138 GW) and between the east and west (94 GW) reducing requirements for investments in storage that would be required otherwise to compensate for regional mismatches between demands for power and the temporal variability in the supplies from wind and solar. Improved

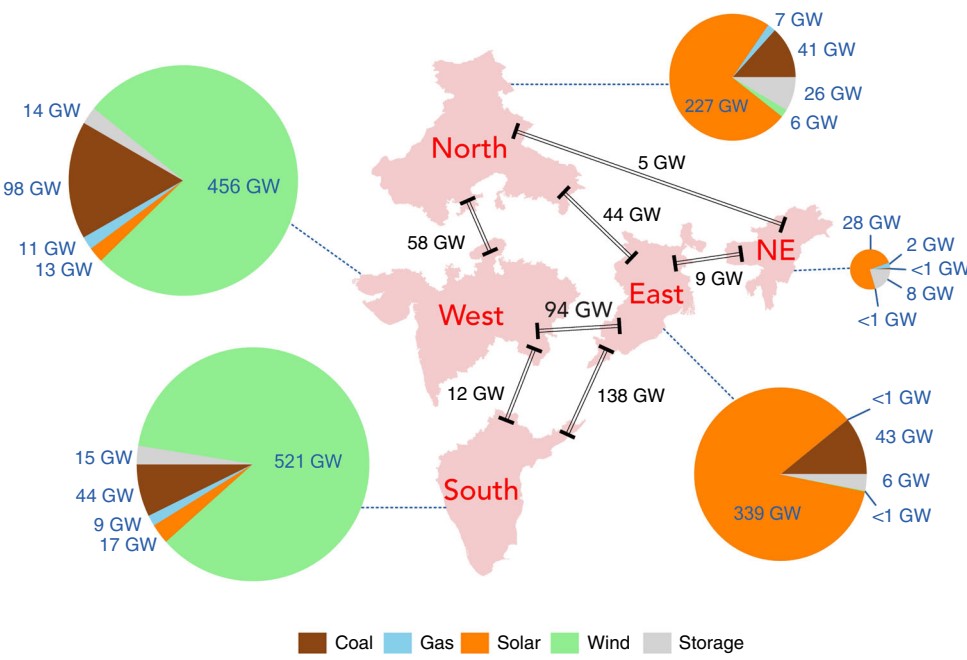

**Fig. 2 Regional capacities.** Regional distribution of capacities inferred for 2040 using the cost-optimization model applied to the standard 80% renewables scenario. Links between regions indicate capacities identified for optimal interregional transmission. The size of the pie charts is proportional to the cumulative generation capacities for each region. This figure was constructed in MATLAB version 2017b and edited in Adobe InDesign 2020.

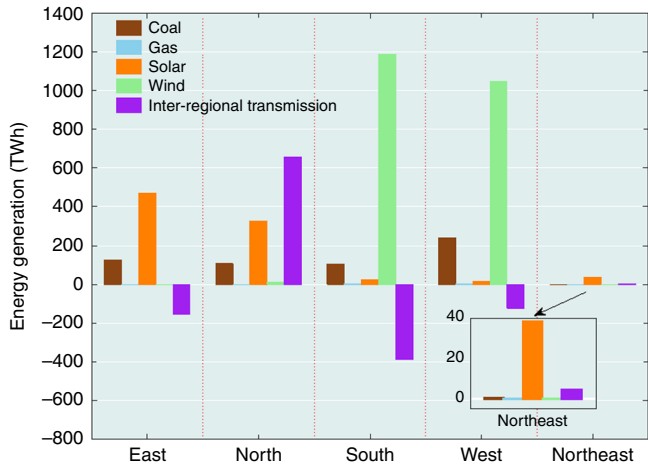

**Fig. 3 Regional energy generation distribution.** Regional distribution of energy generation for different power sources for 2040 inferred using the standard model assuming 80% commitment of renewables. Data identified as interregional transmission refer to interregional transfers of power for the existing and newly expanded transmission network. Negative values identify exports; positive values imports.

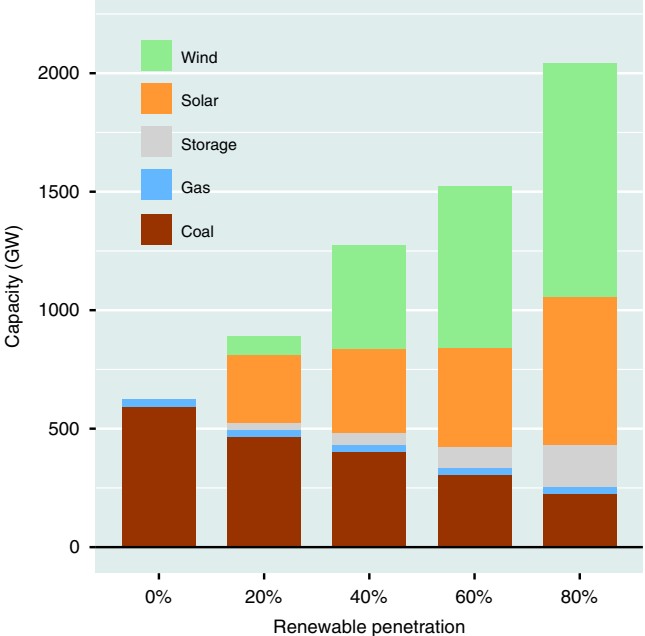

**Fig. 4 National generation mix.** Country-wide generation mix of capacities (GW) for different levels of renewables identified using the optimization model with parameters appropriate for the standard model.

transmission connection between these regions could mitigate the variability of wind and solar power, while reducing at the same time requirements for flexibility resources such as storage.

The contribution of the different sources on an annual basis to the supply of power in different regions is illustrated for the 80% renewables scenario in Fig. 3. Notably, 56% of the power consumed in the north is supplied by transfer from elsewhere. Export, on the other hand, accounts for important fractions of the power produced in the south and east, 39% and 33%, respectively. On a national basis, wind is responsible for 58% of total power consumption, followed by solar (23%) and coal (15%) with minor contributions from hydro (3%), nuclear (1.5%), and gas (<0.2%).

**National results.** The prior section focused specifically on the 80% renewables scenario. This section discusses results obtained assuming different levels of renewables. The optimal assignment of capacities for individual components of the national power system as a function of differences in assumed levels of commitment to renewables is illustrated in Fig. 4. The reference case assumes that the demand for power is met entirely by investments in coal and gas plants complementing existing hydro and nuclear facilities (no renewables). Coal-fired plants are favored in

this instance, responding mainly to the lower cost for coal as compared to gas. The optimization analysis indicates that meeting the 2040 projected demand for power in a cost optimal framework in the absence of a significant contribution from renewables would require construction of as much as 393 GW of coal-fired plants in addition to 34 GW of gas-fired systems. With increasing reliance on renewables, the role of fossil sources, specifically coal, is steadily diminished, decreasing to 227 GW as the contribution from renewables increases from zero to 80%. The commitment to gas remains relatively steady over the entire range of renewables considered here, varying from a low of 28 GW to the high of 34 GW identified with the reference, zero-renewables, scenario. At lower levels of commitment to renewables (less than about 40%), solar is favored relative to wind responding to the lower price (per unit of power) for the former. Wind becomes the investment of choice at higher levels of renewables reflecting the more favorable values of related capacity factors in this case. As indicated, the cost efficiency of the power system is enhanced at all levels of renewables with investments in storage (peaking at 220 GW) and with strategic expansion of the transmission network (peaking at 389 GW), the latter more consequential than the former.

Costs for operation of the power system over the course of a year (2040) for the different levels of renewables considered here are summarized in Fig. 5. The analysis accounts for amortized costs for capital investments in coal, gas, wind, solar, transmission, and storage, in addition to expenses for fuel and operations in the case of the coal and gas-fired systems. Capital costs for coal and gas plants are amortized over a period of 30 years assuming an interest rate of 7% per year[20]. Costs for wind, solar, transmission, and storage are amortized over a shorter interval, 20 years, and assume a similar interest rate. Costs for the reference, zero-renewables case, are dominated by outlays for fuel, primarily coal ($203 billion), adding to a total annual expense of $233 billion. Costs for the 80% renewable case break down as follows: capital investments for wind, solar, and storage, $88.6 billion, $31.0 billion, and $13.4, respectively (operational costs minor by comparison); and costs for capitalization and operation of coal and gas-fired systems, $30.8 billion. Notably, overall costs for the 80% renewables case are less than the costs identified for the zero-renewables reference scenario, $181.8 billion as compared to $233 billion reflecting the fact that energy resources for

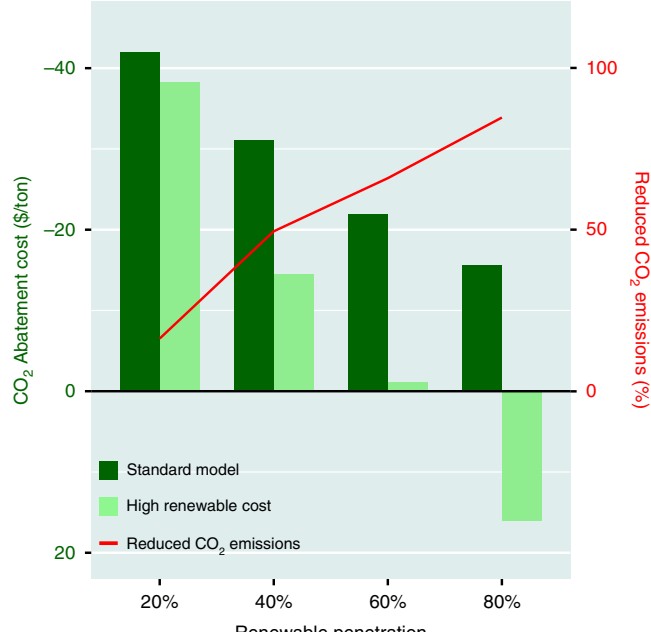

**Fig. 6 CO$_2$ emissions reductions and costs.** Costs for CO$_2$ emission reductions (in $/ton CO$_2$) obtained by comparing the cost per unit CO$_2$ at a given renewable penetration to the 0% renewable scenario. Negative costs denote an option where both system costs and CO$_2$ emissions are reduced. Dark green bars indicate the standard model (low renewable cost) scenario and light green bars represent the high renewable cost scenario. The red line denotes the percent reduction in CO$_2$ emissions associated with different renewable scenarios relative to the 0% renewable reference. For discussion of considerations used to derive costs, see text.

wind and solar are free in contrast to the significant fuel and operational costs associated with NGCC and coal-fired power plants, $0.02/kWh and $0.03/kWh, respectively. If, rather than the low capital costs for wind and solar adopted for purposes of the standard model, we were to consider prices at the higher limit of the possible range, specifically the values detailed in the high renewable cost scenario presented in Supplementary Table 5, the cost for the 80% case would be marginally higher than for the zero-renewables reference, $280 billion compared to $233 billion. Significant investment in storage occurs beyond the 40% renewable level, peaking at 220 GW with 80% renewables, contributing to as much as a 16% cost reduction and a 49% decline in curtailment.

**Implications for CO$_2$ emissions**. Reductions in emissions of CO$_2$ predicted as a function of varying levels of investments in renewables, together with related costs, are summarized in Fig. 6. Relative to the zero-renewables case, reductions in emissions developed using the standard model range from 16% corresponding to the 20% renewables assumption, to more than 85% when the investment in renewables is allowed to rise to 80%. Costs for mitigation of CO$_2$ emissions for any assumed level of renewables are determined by considering first the absolute level of the associated emission reductions referenced with respect to the zero-renewables standard, and second the difference in cost for the power supplied, referenced again to this standard. If the cost for the power generated for a particular choice of renewables is less than the cost for the reference, the overall cost for the reduction in emissions is negative (i.e., it costs less to produce power while still realizing a significant reduction in emissions). Costs are negative over the entire span of scenarios considered here with the standard model, ranging from minus $41.9 per ton

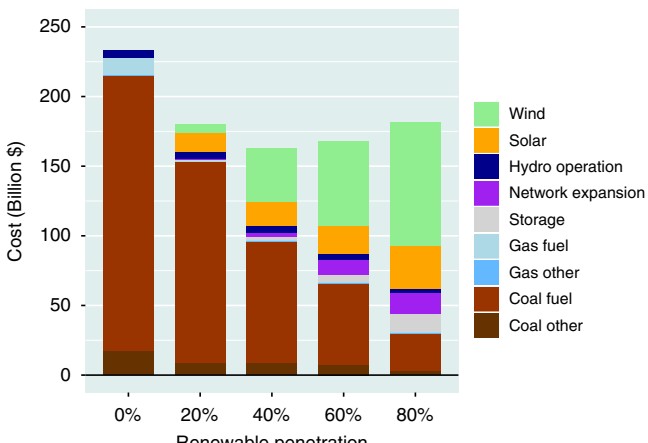

**Fig. 5 National power system costs.** Annual costs for the 2040 power system estimated on the basis of the standard model for different levels of renewables. Coal fuel and gas fuel refer to the costs for fuel for the existing and expanded coal and gas-fired plants. Coal other and gas other refer to the amortized capital and operational and maintenance costs for the new plants.

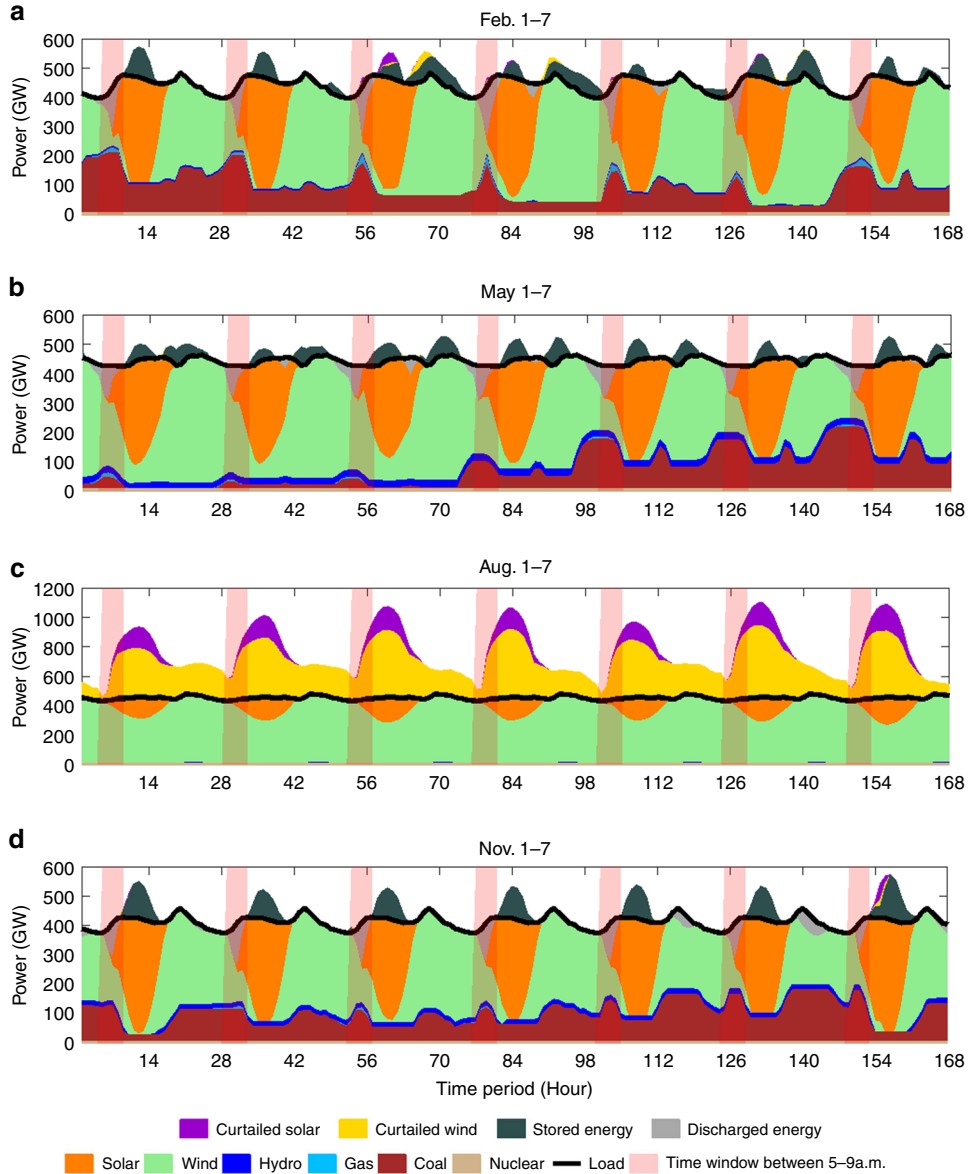

**Fig. 7 Seasonal hourly power balance.** Hourly power balances defined for –seasonally representative weeks on the basis of the 80% renewables standard model simulation: February 1–7 (**a**), May 1–7 (**b**), August 1–7 (**c**), and November 1–7 (**d**).

$CO_2$ in the 20% renewables case to minus $15.6 per ton $CO_2$ in the 80% case. The figure includes also costs for the high wind-high solar price scenario. Costs are marginally less favorable in this instance, varying from negative $38.3 per ton $CO_2$ with 20% renewables, transitioning to positive values for levels of renewables >60%, rising to $15.9 per ton $CO_2$ positive when the penetration of renewables climbs to a level of 80%.

**Seasonal considerations**. Power balances derived for four representative weeks distributed over the four seasons, winter, summer, monsoon, and fall, are presented in Fig. 7. Load demands are indicated by the continuous black curves. Vertical bars are included to identify time windows between 5 am and 9 am. The contributions of power from wind and solar are greatest during the monsoon season as indicated by the data for August 1–7. Significant fractions of the potentially available wind and solar resources, 25% and 26%, respectively, have to be curtailed in this case. Integrated over the entire year, curtailment is less significant, 6.1% overall in the case of wind, 4.3% for solar. Storage

plays an important intermittent role during winter, summer and fall but is unimportant during the high wind-high solar monsoon season. Power captured by storage systems is deployed often during the early morning hours when the supply from solar PV has not yet reached its peak mid-day potential. Coal provides a significant source of baseload power, particularly in fall and winter, but is notably absent during the monsoon period.

## Discussion

The objective of this study was to identify prospects for a significant contribution of renewable sources, specifically wind and solar, to India's electricity future, including opportunities and costs for related reductions in emissions of $CO_2$ by the year 2040. We applied a novel optimization model to identify least cost options to meet India's projected future power demand while allowing for coordination between power system planning and operation as well as expansion of the interregional transmission network. The striking conclusion is that investments in renewables could play an important role in reducing the overall costs

for delivery of electricity in India's future power system. In particular, assuming a commitment to 80% renewables using the standard model, the overall costs for delivery of power in 2040 were projected to be lower by as much as $50 billion compared with expenses for the 0% renewable reference case, equivalent to a decline of 0.5 cent/kWh for retail electricity prices. Emissions of $CO_2$ are reduced by as much as 85% with renewables at 80%, a potential cut in emissions of 3.4 Gt $CO_2$ per year, slightly higher than the level of emissions, 2.5 Gt $CO_2$, that applied in 2017. The findings summarized here provide clear implications for energy and environmental policies for India's electricity sector in planning for a low-carbon 2040.

There are a few caveats that should be noted. First, we considered technological improvements for renewables only in terms of cost; production from renewables due to improved design could also yield greater electricity output. Given the steady growth over the past few decades in capacity factors and overall capacities for renewables, it is possible that there could be significant improvements in these numbers by 2040. Projections for these technological improvements are, however, beyond the scope of this paper.

Second, we note that output from wind turbines and solar panels are weather-dependent. A number of studies [e.g., refs. [21,22]] have shown that India may be undergoing a steady decline in wind speeds, particularly over the monsoon season, which could have bearing on future power output from renewables. In addition, changes in the optical properties of the atmosphere due to aerosol emissions could play a significant role in modulating solar irradiance over the country. Future work will address these concerns by studying implications for changes in India's climate associated with changes in aerosol and greenhouse gas emissions.

Third, we did not consider the feasibility of the timescales associated with the implementation of the proposed additional capacity from renewables. Since 2010, India has installed ~25 GW of wind power and 35 GW of solar PV. The 80% renewables scenario studied here would require addition of approximately 1000 GW of wind power and 500 GW of solar PV by 2040. This growth would call for a fundamental shift in India's energy system. Such an adjustment would be demanding but not impossible; for perspective, the Chinese government installed over 750 GW of renewable generation capacity (wind, solar and hydro) over the past decade with ambition to reach 1000 GW of installed wind power by 2050[23]. India lies at a fork in the road in terms of the trajectory of its energy system. Solar and more specifically wind can make important contributions to India's future demand for electricity. These options, we argue here, are not only economically more favorable but also environmentally more constructive, providing opportunities to avoid the continuing reliance on polluting coal envisaged in the country's current long-term energy plan (70% of power from coal by 2040). This pivotal moment requires rigorous assessment of the options available going forward. Supporting measures, such as interregional transmission and storage, are also of importance to guarantee a smooth integration of the variable power sources. The analysis presented here offers a critique of the economic and environmental implications of these options from a perspective covering the next several decades.

Fourth, a complementary pathway for a cost- and energy-effective integration of renewables might be to promote investments in concentrated solar power (CSP) and hydrogen fuel cells, which could reduce the variability of the output from solar resources and could provide for flexible and economically viable opportunities for applications of zero carbon sources in other segments of the Indian economy.

Fifth, the foregoing analysis assumed that the investments considered here would be instituted de novo in 2040. In practice, the more likely scenario would involve a gradual build-up. Overall costs could be somewhat higher in this case, assuming an inability to take full advantage of longer-term projected price declines. Prospects for a more gradual, but still cost-effective, transition to a high renewables future will be explored in more detail in a follow up study. We concede that even a gradual build-up of the renewable capacity proposed here (over 1500 GW in the 80% renewable scenario) is certainly ambitious. We would note, however, the tenfold increase in wind and solar PV capacity realized over the last decade in China and would point further to the report by the Climate Policy Initiative[24] which concluded that India could integrate as much as 390 GW of low-cost wind and solar power by 2030.

## Methods

**Data overview**. The wind data used in the study were derived based on MERRA-2[17], a NASA reanalysis product publicly available in NASA's Goddard Earth Sciences Data and Information Services Center. This database defines hourly wind speeds with a spatial resolution of 0.50° longitude by 0.67° latitude from 1980 to present. Wind speeds at 100 m were extrapolated from 10 to 50 m using the vertical profile of the power law described by Archer and Jacobson[25]. The friction coefficient in the analysis was evaluated using wind speeds represented at 10 and 50 m for each grid cell, as in Lu et al.[26]. Wind power was computed on an hourly basis using the power curve for the MHI Vestas Offshore V164-8.0 MW wind turbine, a typical system employed currently for offshore applications, and the Goldwind 2.5 GW for onshore. Specifications for each technology are summarized in Supplementary Table 2. Capacity factors (CFs), defined by the ratio of electricity generated by a solar installation relative to the realization of its full capacity over the same period, were evaluated on an hourly basis at the spatial resolution of the NASA database. The solar data used in the study were derived from NASA's GEOS-5 FP database[27], which identifies hourly temperatures and incident solar radiation at a spatial resolution of 0.25° latitude by 0.31° longitude. We employed an integrated solar PV assessment model in evaluating the performance of solar PV systems, following the approach described by Chen et al.[18]. The spatial variation of factors impacting CF were modeled consistently, accounting for tilt, packing density, sun shading, and temperature. Hourly solar power values were calculated assuming installation of fixed-tilt polysilicon PV modules with a 16.2% conversion efficiency.

**Onshore filter**. Onshore areas that are forested, urban, or covered with water or ice were filtered according to data from the NASA MODIS (Moderate Resolution Imaging Spectroradiometer) satellite MCD12C1 dataset[28]. Slope data were derived from the Shuttle Radar Topography Mission (SRTM) Global Enhanced Slope Database[29] with a spatial resolution of 1 arc-s (~30 m). Grids characterized by slopes of more than 20% or by heights of more than 3000 m were excluded as inappropriate for deployment of onshore wind power systems.

**Offshore filter**. To determine locations suitable for offshore wind in India, we filtered data spatially based on a number of criteria. First, only locations within India's Exclusive Economic Zone (EEZ) were considered. India's boundaries for the EEZ were taken from Marine Regions, a database which aggregates information from a number of regional and national providers[30]. Another filter adopted was to consider only fixed-bottom turbines, which require offshore depths of less than or equal to 60 m. The offshore depth data used here were taken from the General Bathymetric Chart of the Oceans (GEBCO) One Minute Grid, a global bathymetric grid providing data at a one-arcminute resolution[31]. Finally, we removed areas from each grid according to environments designated as either "Special Marine Reserves" (environmentally-protected regions) or shipping routes. Areas for the Special Marine Reserves are defined in ref. [32]. The $SO_2$ emissions compilation from MERRA-2 was used as a surrogate in the identification of shipping routes, and 20% of a cell's area was removed for locations defined as emitting $SO_2$ at a rate higher than $10^{-11} \, \mathrm{kg \, m^{-2} \, s^{-1}}$.

**Solar filter**. This study used slope, land use type, and solar radiation as criteria to identify areas suitable for solar farm development, following the approach described by Chen et al.[18]. The maximum permissible slope was set at 5%. As with onshore wind, the SRTM database was used to calculate terrain elevation and slopes for each grid. Suitability factors were selected according to land use types with higher values allocated to land areas with sparse vegetation and low ecological productivity[33] (Supplementary Table 6). The MODIS data were used to filter unsuitable land areas from this analysis, excluding forests, water bodies, permanent wetlands, croplands, cropland/natural vegetation mosaic, and snow and ice environments (land classifications are indicated in Supplementary Fig. 4). Areas excluded by these filters were assigned 0% as suitability factors. For exploitable

areas, suitability factors ranging from 5 to 20% were assigned to each land use type. The minimum solar radiation required for exploitable land areas was set at 1400 kWh/(m²·a), a typical threshold value for acceptable solar resources[18]. And, it should be noted, the current study does not allow for a potential source of carbon-free electric power from solar panels installed on roof tops, a development that could be facilitated by appropriately targeted policy initiatives.

**Regional power capacity.** Musial et al.[34] estimate that the spacing appropriate to minimize turbine-turbine interference for offshore wind is equivalent to ~7 rotor diameters, corresponding to a deployment density for turbines of one per 1.04 km². The area for each latitude/longitude grid cell was divided by this value to compute the number of turbines that could fit maximally into a given cell. It should be noted that this spacing does not account for the downstream wake effect, which is of too small scale to be modeled accurately using the MERRA-2 data. Given that the average downstream power loss is on the order of 5%[35], the wake effect should not have a significant bearing on the present results. The potential installed capacity (in GW) is computed by multiplying the number of turbines in a cell by the turbine power (8 MW in this case). Onshore power is calculated similarly, using a spacing of 9 rotor diameters (one turbine per 0.64 km²) and turbine power of 2.5 MW. The solar power PV capacity potential (in GW) is defined by the packing factor obtained by multiplying the power per unit area of the PV panels (161.9 Wm⁻²) by the area available for their placement (factoring in solar filter constraints as described before). The spatialized packing factor here refers to the effective panel area per square meter of land area, which is determined by the solar PV tilt, azimuth angle (east-west orientation), and the spacing between neighboring PV panel footprints. The tilt setting assumed in the study follows the method proposed by Jacobson[36], and the orientation of the panels was set to face the equator. The principle to determine the spacing between footprints is to ensure that minimal shading will occur for most of sunlight hours throughout the year. The spacing was calculated using the solar altitude angle for 3 PM at the winter solstice, the day for which shading is likely to be most significant.

**Power generation.** The next step is to quantify the power that could be supplied to individual regions. For offshore wind, we assumed that the wind resource available over a given location in India's EEZ was under the jurisdiction of the country's nearest region. These regional divisions, along with the mean of on- and offshore wind and solar PV CFs over the year 2016 are indicated in Fig. 2. From the installed capacity and CF data, estimates of available energy for each technology $E$ (lat,lon,t) (in kWh) were computed using the equation:

$$E(lat, lon, t) = CF(lat, lon, t) \times C(lat, lon) \times 8760 \qquad (1)$$

where $C(lat,lon)$ represents the installed capacity at a given location, $CF(lat,lon,t)$ is the CF at the location and 8760 defines the number of hours in a year.

**Projected costs for each technology.** The globally averaged price for PV decreased from $4.60/W in 2010 to $1.20/W in 2018[37]. Pachouri et al.[3] argued that these prices should continue to decline, projecting a decrease of 3% per year to 2024, 2% per year from 2024 to 2027, and 1% per year thereafter. For present purposes, we assume a range of prices for PV panels in 2040 varying from a low of $0.55/W to a high of $1.65/W. All of the costs quoted here are defined in terms of 2018 US dollars. Prices for onshore wind have also declined, dropping from $1.90/W in 2010 to $1.20/W in 2018[37]. Pachouri et al.[3] projected a more modest decrease for future prices in this case, 1% per year. For present purposes, we consider a range of costs for 2040 onshore wind installations varying from $0.98/W to $1.95/W, with higher prices, $1.30/W to $2.30/W, assigned for offshore facilities. Current trends would appear to favor the lower of the costs quoted here for all three applications. Accordingly, we elected to emphasize for purposes of the standard model in what follows the lower of the ranges of values indicated here ($0.55/W for solar PV, $0.98/W for onshore and $1.30/W for offshore wind)[37]. The low-cost projections for these renewables are consistent with cost estimates from NREL[38]. The sensitivity of results to the choice of costs will be discussed later and more extensively in the SI.

India has abundant reserves of coal, fourth largest in the world trailing only the US, Russia and China. The contemporary price for coal in India averages about $3.5/MMBTU[39]. We assume that this price is unlikely to change much by 2040 and adopt accordingly a reference future cost for coal of $3.6/MMBTU. It is more difficult to predict future prices for gas, given the sensitivity of prices for this commodity to vagaries of the international market. NITI Aayog and IEEI[39] suggest a range for future prices from $6/MMBTU to $15/MMBTU. For present purposes, we adopt a value of $7.65/MMBTU, near the midpoint of this range. The sensitivity of conclusions to this choice will be discussed in what follows.

As indicated earlier, in seeking the least cost strategy to minimize future costs for electricity while organizing a significant shift from coal to wind and solar, we propose to allow for cost-effective expansions of the interregional transmission grid in addition to investments in storage. Estimated costs for expansion of the interregional transmission grid are summarized in Supplementary Table 2. These costs were defined by considering expenditures per unit of power for investments involved in development of the current grid[40]. The higher costs associated with

specific interconnections reflect primarily the greater distances involved with these links.

**Options for storage.** A variety of options are available for storage of power. Mechanical systems include pumped hydro, compressed air, and flywheels. Chemical options refer mainly to batteries. Two considerations are involved in assigning relevant costs: the peak power capacity of the system (measured for example in kW), and the capacity of the system to store energy (measured specifically in kWh). A range of prices for different systems, adopted from Safaei and Keith[41], is presented in Supplementary Table 3. For purposes of the standard model, we select the option identified as Medium Cost. The optimization model described below is charged with exploring the least cost option for any particular application, recognizing the distinctions between the power and energy capabilities of individual systems. Pumped hydro is responsible for the bulk of the 140 GW of power storage currently deployed globally and is likely to play an important role in the future also for India. Capital expenditures for construction of pumped hydro facilities are high, however, relative to costs for batteries[42]. Responding to the disparity in prices for capital investments in pumped hydro versus batteries, the current analysis concludes that batteries are likely to provide the option of choice for storage of power for India at least over the time interval considered here.

**Optimization model for India's energy system and its capacity expansion.** The generation and transmission capacity expansion results for different levels of renewables were obtained based on a capacity expansion model optimizing jointly investment decisions and hourly system operations accounting for a full set of flexibility constraints. The model allows for potential deployment of defined renewable resources, for thermal generation, for energy storage and for upgrades in interregional transmission.

The decision variables for the energy system capacity expansion model (ESCEM) involve two components. For capacity investments, the decision variables account for invested capacities for each type of generation technology in each region, the capacity of storage deployed, and the capacity for transmission between different regions. For system operation, the decision variables allow for the available capacity and for the hourly dispatched output for each category of generation and storage for each region. The capacity available during the dispatch phase is interlinked with the investment decisions.

The objective of the ESCEM is to minimize the overall system cost, which includes two parts: (1) system annual operational costs, the sum of hourly fuel costs, start-up costs and operational costs for storage, thermal power, hydropower and nuclear power systems; and (2) amortized capacity investment costs, fixed O&M expenses and costs for the interregional network expansion.

The model considers a full set of constraints for the system operation. Hourly power balance as well as reserve constraints are incorporated for each region. Flexibility constraints for thermal units are also included with maximum and minimum generation limits defined, and with specification of ramping and minimum on/off time constraints. Operational constraints relating to energy storage are also considered based on different characteristics of storage technologies. Limitations on interregional power flow are incorporated in optimizing regional power exchange. Finally, renewable portfolio requirements are incorporated as an additional constraint.

To accelerate the calculation at such large scale, a novel flexibility method described in ref. [43] is employed to reduce the modeling complexity and improve the computational efficiency. The Units are grouped in the model with similar operational characteristics (same fuel type, similar nameplate capacity) to be dispatched based on aggregated power generation. There are six groups (categories) for each of the five regions in India and the total online capacity at each time interval is calculated by a combination of on-off status for all individual units in the group.

The mathematical formulation of the proposed optimization model is detailed further in the Supplementary Method and validation of the model is discussed in Supplementary Note 2.

**Projecting India's energy system in 2040.** The proposed ESCEM accounts both for the expansion projected in power demand and the annual hourly operation for India's energy system in 2040 covering five regions (East, North, South, West and Northeast), accounting for regionally distributed load, power plants (thermal, hydro, and nuclear), renewables (solar and onshore/offshore wind), possible energy storage systems, and exchanges for the interregional power grid. A detailed description of the energy system configuration in India is presented in the SI. Following results from Spencer et al.[44], we scale the hourly power demand to 2040 for each region assuming a compound annual growth rate of 6.5%, which accounts for population and economic growth. Estimates of the hourly demand in 2016 were obtained from POSOCO[16], and are indicated in Supplementary Fig. 5.

To project the role of coal in India's future energy system, we consider technological improvements for coal-fired units in 2040 motivated by India's Ministry of Power proposal to renovate and replace inefficient coal-fired units. Details of the updated configuration can be found in Supplementary Table 3. Other unit information (gas, hydro, and nuclear) is similarly derived from Ministry of Power proposals. Nuclear units are fixed; i.e., power output is time independent

because the plant must be operational all of the time. Because of the relatively low cost and high flexibility of hydropower plants, we assume a fixed capacity of 55 GW based on the Indian government's future investment plan for hydropower[45]. Locations for coal, gas and nuclear plants are indicated in Supplementary Fig. 3.

Costs for renewables are projected to decline significantly in the future, in response to technological improvements and benefits from learning experience. We consider fixed cost reduction rates of 15% for onshore, 35% for offshore, and 45% for solar PV from present-day values to 2040. To cover the most conservative and optimistic estimates for renewable investment costs, we consider two scenarios: a low-cost scenario based on the lowest renewable investment costs ($975 kW$^{-1}$ for onshore wind, $1300 kW$^{-1}$ for offshore wind, and $550 kW$^{-1}$ for solar PV) and a high-cost scenario based on the highest renewable investment costs ($1955 kW$^{-1}$ for onshore wind, $2300 kW$^{-1}$ for offshore wind, and $1650 kW$^{-1}$ for solar PV), following analyses of current renewable systems from IRENA[37]. Results derived from other future pathways are indicated in Supplementary Figs. 6–10.

As an important operational component of India's energy system, capacity expansion for seven interregional transmission corridors (indicated in Fig. 2) is considered in the model. We assume that the number and location of corridors are fixed and the transmission capacity for a given corridor can be expanded according to an expansion factor, defined as the ratio of total current investment cost for a corridor to its capacity. The calculation process and expansion factors are presented in the Supplementary Method and Supplementary Table 2, respectively.

Total sector emissions of $CO_2$ are aggregated from hourly emissions of all thermal generators. Emissions factors of $CO_2$ for coal and gas-fired units are derived from the Ministry of Power[46]. Detailed description of emission factors and calculations of emissions for different scenarios are indicated in the Supplementary Note.

## Data availability
The cost breakdown, $CO_2$ status and generation mix of the country-wide energy system for scenarios investigated in the cost-optimization model are available in Supplementary Data 1.

## Code availability
Code for the model can be made available upon request.

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

## Acknowledgements

T.L., P.S., and M.M. are supported by the Harvard Global Institute. X.C. is supported by National Science Foundation China (51907066) and State Key Laboratory on Smart Grid Protection and Operation Control of NARI Group, through the open topic project (20171613). We acknowledge instructive conversations with Prem Shankar Jha and Chris Nielsen.

## Author contributions

T.L., P.S., X.C., S.C., X.L., and M.M. contributed equally to the genesis and conduct of this research and to the writing of the resulting manuscript.

## Competing interests

The authors declare no competing interests.
