## [Peer Review File · Nature Communications]

Reviewers' comments:

Reviewer #1 (Remarks to the Author):

This manuscript assesses the feasibility of high-levels of renewables integration in the Indian energy system. The renewables considered here are only wind and solar. The literature review clearly emphasises the need for decarbonisation of the Indian power market, which is currently highly dependent on fossil fuels. The methodology used in this work seems to be sound, although no validation/verification of the model outputs was performed. The objective of this work is clearly defined. However, in my view, the novelty of this work is not sufficient. There are also some inconsistencies in the results. Therefore, I cannot recommend the publication of this manuscript in Nature Communications in the current form. Please find my detailed comments and suggestions below.

Abstract

Lines 17-18: The authors state that "[...] renewables, wind and solar, could be responsible for as much as 80% of anticipated 2040 power demand [...]". It is not clear how renewables can represent the demand side of the economy. I would expect that renewables "account for 80% of power supply" or "meet 80% of power demand".

Introduction

Page 2: The authors use a factor of 109 rather than prefix "giga" to quantify the CO₂ emissions and use the prefix "giga" rather than a factor of 109 to quantify the capacity of Indian power generation. I suggest using the prefix "giga" in both cases to maintain consistency throughout the manuscript.

Page 2: The Indian power generation was 344 GW in 2018, and just 65.7 GW were jointly generated by solar and wind. Considering that India is a rapidly growing economy, its power demand will increase significantly by 2040. I wonder whether achieving 80% power supply from renewables, considering their low availability, is reasonably achievable?

Page 3: The authors benchmark the scenarios with high penetration levels of renewables with the conventional thermal generators and claim that wind/solar can provide "a cost competitive alternative to India's current coal dominated power future [...]". Have the authors considered other decarbonisation strategies for the power sector, such as the large-scale use of sustainable biomass or CO₂ capture? It will be useful to consider different decarbonisation scenarios before making a claim which one is the most promising from the cost perspective.

Page 6: The PV price of \$0.55/W seems low, even in 2040. Has this value been adjusted for the effect of inflation?

Pages 12-13: In Figure 4, it is clear that a significant overcapacity of renewables (roughly 3x that of power generation based on fossil fuels) is required to meet the energy demand in 2040. With the current prices of ~\$0.5/W for advanced NGCCs and ~\$1/W for advanced coal-fired power plants, it is somewhat unclear why implementation of a much larger amount of renewables capacity at comparable price decreases the total cost of the network. It is unclear if the economic model accounts for the carbon tax benefits.

Pages 17-19: The discussion section focuses on the main assumptions, limitations of this work and recommendations for future work. I recommend the focus is placed on the potential impact of the outcomes of this work, instead of providing an overview of the main assumptions.

Reviewer #2 (Remarks to the Author):

The paper is interesting and timely. It needs some revisions before publication.

General comments:

1) A thorough review of language is needed.

2) The research gap is not very clearly established. Are there any other studies that address the same aspect for India? What is the novelty of your work? Clearly highlight the research gap and the motivation.

3) Are there any government policies that affect the energy planning decision? These need to be considered in the analysis.

4) I feel that if the background section preceded the model formulation, the flow of the paper would be clearer. In addition, the model formulation section (which seems to represent the methodology in the main body of the text) needs to be divided into sub-sections for clarity.

5) Some of the information, especially the statistics, provided under the model formulation section would be better suited in the background. In addition, some of the results are also included under the model formulation.

6) The economic implications need to be further highlighted in the discussion. What does this mean for the energy costs incurred by the consumers at present?

Page 2:

Ln 30-41: Perhaps this information could be presented in graphical form for further clarity.

Page 3:

Ln 64-65: The sentence is not clear. Please revise.

Page 10

Ln 233-237: What do these results signify?

Response to Anonymous Reviewers

Review 1

This manuscript assesses the feasibility of high-levels of renewables integration in the Indians energy system. The renewables considered here are only wind and solar. The literature review clearly emphasises the need for decarbonisation of the Indian power market, which is currently highly dependent on fossil fuels. The methodology used in this work seems to be sound, although no validation/verification of the model outputs was performed. The objective of this work is clearly defined. However, in my view, the novelty of this work is not sufficient. There are also some inconsistencies in the results. Therefore, I cannot recommend the publication of this manuscript in Nature Communications in the current form. Please find my detailed comments and suggestions below.

Reply:

- Thanks for the detailed review and constructive comments for this paper. We have improved the model, results and discussion sections following your suggestions, with changes colored in red. The inconsistency issue on units has been fixed. Several confusing parts have been clarified.
- The research gap has now been more clearly described. The major contributions of the paper lie in the following two respects:
 - On lines 70-83: “The primary challenge in planning for power systems under high levels of renewables is to reconcile the conflict between the variability of renewable sources and the intrinsic inflexibility of thermal power systems. This has led to a loss of more than \$10 billion for China due to curtailments of wind and solar power. Important contributions have been conducted for national level pathway studies [R1-R6]. However, the flexibility issue has been largely simplified, resulting in over optimistic projections of carbon abatement costs and under investments in flexible power generating resources. Here we introduce an integrated renewable energy system planning model designed to co-optimize investments for generation, transmission and storage expansion with detailed treatment of system operations considering not only requirements for balancing supply and demand, but also hourly ramping, reserves, minimal load, and timing involved in start-up and shut down of thermal units. To accelerate the optimization process, our previously developed Fast Unit Commitment model is applied to reduce the computational complexity for operational simulation.”
 - On lines 332-340: “The striking conclusion is that investments in renewables could play an important role in reducing the overall costs for delivery of electricity in India’s future power system. In particular, assuming a commitment to 80% renewables using the standard model, the overall costs for delivery of power in 2040 were projected to be lower by as much as \$50 billion compared with expenses for the 0% renewable reference case, equivalent to a decline of 0.5 cent/kWh for retail electricity prices.”
- To validate the generation/transmission expansion model developed here, we present two aspects of validation summarized here and elaborated in greater detail in Supplementary Note 2:
 - **The accuracy and computational efficiency of the accelerated operational**

simulation model (FUC model): The core of proposed generation/transmission expansion model is the accelerated operational simulation model, called the Fast Unit Commitment (FUC) model. The FUC model groups identical thermal units and reformulates all flexibility constraints to accelerate the calculation. A thorough comparison between FUC model and conventional unit commitment model was included systematically in our previous publication [R7]. Based on standard test system and an actual regional power system in China, the FUC model could reduce the computational time by a factor of 2000, with less than a 2% deviation of results. These data are indicated in Supplementary Table 7.

- **The effectiveness of the proposed optimal generation/transmission investment model:** One of the major contributions from the modeling perspective is a proposed integrated planning model. Incorporating the FUC model described above, the investment model can consider detailed operational flexibility constraints at the systems to demonstrate the effectiveness for the proposed investment model. We have added a detailed comparison based on Indian energy systems to demonstrate the effectiveness of the proposed investment model. Results show that with our model, renewable curtailment could be estimated more accurately (as indicated in Supplementary Figure 11, reproduced below), and the integrated FUC technology of operational modeling for flexible resources in the generation mix can lower overall costs (as indicated in Supplementary Table 8).

Supplementary Figure 11. The hourly power balance for India over a typical week derived from conventional planning model (PLn-no-flex) and the proposed planning model (PLn-flex), as compared with hourly UC simulation results. (a) PLn-no-flex. (b) Simulation-no-flex: hourly simulation results using UC model for the same time period based on generation mix derived from PLn-no-flex (c) PLn-flex. (d) Simulation-flex: hourly simulation results using UC model for the same time period based on generation mix derived from PLn-FUC.

Abstract

Lines 17-18: The authors state that “[...] renewables, wind and solar, could be responsible for as much as 80% of anticipated 2040 power demand [...]”. It is not clear how renewables can represent the demand side of the economy. I would expect that renewables “account for 80% of power supply” or “meet 80% of power demand”.

Reply:

- Thanks for the comment. We have modified this sentence according to the suggestion for clarity.

Introduction

Page 2: The authors use a factor of 109 rather than prefix “giga” to quantify the CO₂ emissions and use the prefix “giga” rather than a factor of 109 to quantify the capacity of Indian power generation. I suggest using the prefix “giga” in both cases to maintain consistency throughout the manuscript.

Reply:

- Thanks for the suggestion. We have now used the prefix “giga” to quantify both the capacity of power generation and emissions.

Page 2: The Indian power generation was 344 GW in 2018, and just 65.7 GW were jointly generated by solar and wind. Considering that India is a rapidly growing economy, its power demand will increase significantly by 2040. I wonder whether achieving 80% power supply from renewables, considering their low availability, is reasonably achievable?

Reply:

- The physical potential for wind and solar in India exceeds 30,000 GW, and thus an abundant resource is potentially available for large-scale deployment to supply the projected peak-hour power demand of 627 GW in 2040 [R8].
- We reference the situation in China, whose total capacity for renewable resources reached 750 GW by the end of 2019, largely due to investments in the past decade. With rapidly decreasing costs, the levelized cost of electricity of wind and solar could be competitive or even lower than fossil units in the near future. Thus we anticipate that the pace for renewable investments could certainly accelerate rapidly in the next 20 years. In this context, deploying renewable resources at such scale is achievable over a time period of two decades.
- The real challenge lies in the accommodation of renewables in an intrinsically inflexible power system such as that in India. Deployment of storage technologies, flexible thermal units, and interregional transmission lines to mitigate variability at broader geographical scope would be critical to avoid curtailments and to support sustained renewable investments.
- We thank the reviewer for raising the question. These points and associated changes are reflected in the revised introduction and discussion sections.

Page 3: The authors benchmark the scenarios with high penetration levels of renewables with the conventional thermal generators and claim that wind/solar can provide “a cost competitive alternative to India’s current coal dominated power future [...]”. Have the authors considered other

decarbonisation strategies for the power sector, such as the large-scale use of sustainable biomass or CO₂ capture? It will be useful to consider different decarbonisation scenarios before making a claim which one is the most promising from the cost perspective.

Reply:

- Thanks for the insightful comments. We recognize that there are other options that might contribute to a lower carbon future for India including increased reliance on hydro, nuclear and potentially biomass, in addition to targeted investments to improve energy efficiency. We choose to focus here on wind and solar recognizing the emphasis that has been placed on these resources most recently by the Indian government [R8]. The government plan largely reflects the rapid decline in costs for wind, solar and storage, and renewables will soon be competitive even with conventional coal-fired power plants. We have added relevant discussion in the updated manuscript on lines 100-107.

Page 6: The PV price of \$0.55/W seems low, even in 2040. Has this value been adjusted for the effect of inflation?

Reply:

- We thank the reviewer for raising this question. All prices in this study have been expressed in 2018 USD, meaning that we have ignored the effect of inflation. The PV price here refers to the report [R10], predicting future costs with a range of \$550-1650/kW. Here we assume the PV price is \$550/kW and \$1650/kW in the low cost and high cost scenarios respectively to reflect the uncertainty. Notably, the low-cost projections for these renewables are consistent with cost estimates from NREL [R11].

Pages 12-13: In Figure 4, it is clear that a significant overcapacity of renewables (roughly 3x that of power generation based on fossil fuels) is required to meet the energy demand in 2040. With the current prices of ~\$0.5/W for advanced NGCCs and ~\$1/W for advanced coal-fired power plants, it is somewhat unclear why implementation of a much larger amount of renewables capacity at comparable price decreases the total cost of the network. It is unclear if the economic model accounts for the carbon tax benefits.

Reply:

- The overall cost of a generation technology includes two major components: the capital cost (\$/W) and the operational cost (\$/Wh). The operational (fuel) cost for NGCCs and coal-fired power plants are \$0.02/kWh and \$0.03/kWh, respectively [R8], while energy resources for wind and solar are free. Therefore, introduction of renewables to meet 80% of anticipated 2040 power demand can significantly reduce the annual operational cost for thermal power plants by \$153 billion, leading to a more cost-effective pathway as compared to the zero renewables case. This description is presented now in lines 257-259 and Figure 5.

Pages 17-19: The discussion section focuses on the main assumptions, limitations of this work and recommendations for future work. I recommend the focus is placed on the potential impact of the outcomes of this work, instead of providing an overview of the main assumptions.

Reply:

- Thanks for the suggestion. We have adjusted the discussion section accordingly, focusing more specifically on potential impacts of this work, described as follows:
- On lines 329-338: “We applied a novel optimization model to identify least cost options to meet India’s projected future power demand while allowing for coordination between power system planning and operation as well as expansion of the inter-regional transmission network. The striking conclusion is that investments in renewables could play an important role in reducing the overall costs for delivery of electricity in India’s future power system. In particular, assuming a commitment to 80% renewables using the standard model, the overall costs for delivery of power in 2040 were projected to be lower by as much as \$50 billion compared with expenses for the 0% renewable reference case, equivalent to a decline of 0.5 cent/kWh for retail electricity prices. Emissions of CO₂ are reduced by as much as 85% with renewables at 80%, a potential cut in emissions of 3.4 Gt CO₂ per year, slightly higher than the level of emissions, 2.5 Gt CO₂, that applied in 2017.”
 - On lines 364-374: “India lies at a fork in the road in terms of the trajectory of its energy system. Solar and more specifically wind can make important contributions to India’s future demand for electricity. These options, we argue here, are not only economically more favorable but also environmentally more constructive, providing opportunities to avoid the continuing reliance on polluting coal envisaged in the country’s current long-term energy plan (70% of power from coal by 2040). This pivotal moment requires rigorous assessment of the options available going forward. Supporting measures, such as interregional transmission and storage, are also of importance to guarantee a smooth integration of the variable power sources. The analysis presented here offers a critique of the economic and environmental implications of these options from a perspective covering the next several decades.”

Again, we thank the reviewer for the considerate comments. We hope our revisions address all of the concerns raised.

References

- R1 Shukla, P. R. et al. Pathways to deep decarbonization in India, SDSN - IDDRI. (2015).
- R2 Gadre, R. & Anandarajah, G. Assessing the evolution of India’s power sector to 2050 under different CO₂ emissions rights allocation schemes. *Energy Sustain Dev*, 50, 126–138 (2019).
- R3 Tokimatsu, K., Yasuoka, R., & Nishio, M. Global zero emissions scenarios: The role of biomass energy with carbon capture and storage by forested land use. *App. Energy*, 185, 1899–1906 (2016).
- R4 Grubler, A., et al. A low energy demand scenario for meeting the 1.5C target and sustainable development goals without negative emissions technologies. *Nature Energy*, 3, 515–527 (2018).
- R5 Anandarajah, G. & Gambhir, A. India’s CO₂ emission pathways to 2050: what role can renewables play? *Appl. Energy*, 131, 79–86 (2014).
- R6 Mittal, S., Dai, H., Fujimori, S., & Masui, T. Bridging greenhouse gas emissions and renewable energy deployment target: comparative assessment of China and India. *Appl. Energy*, 166, 301–313 (2016).

R7 Han, X., Chen, X., McElroy, M. B., Liao, S., Nielsen, C. P., & Wen, J. (2019). Modeling formulation and validation for accelerated simulation and flexibility assessment on large scale power systems under higher renewable penetrations. *Applied energy*, 237, 145-154.

R8 Pachouri, R., Spencer, T., & Renjith, G. Exploring electricity: Supply-mix scenarios to 2030. (The Energy and Resources Institute, 2019).

R9 Sharma, N., Singh, U., & Mahapatra, S. S. (2019). Prediction of cost and emission from Indian coal-fired power plants with CO₂ capture and storage using artificial intelligence techniques. *Frontiers in Energy*, 13(1), 149-162.

R10 IRENA. Renewable Power Generation Costs in 2017, International Renewable Energy Agency 2018. <https://www.irena.org/publications/2018/Jan/Renewable-power-generation-costs-in-2017> (accessed March 21, 2019).

R11 NREL. 2019 Annual Technology Baseline. Golden, CO: National Renewable Energy Laboratory. <https://atb.nrel.gov/electricity/2019>. (2019).

Reviewer: 2

Reviewer comments to the author:

The paper is interesting and timely. It needs some revisions before publication.

Reply:

- Thanks for the insightful comments and constructive suggestions. We have modified the paper accordingly, colored in red. Detailed replies to each of the questions raised are presented below.

1. A thorough review of language is needed.

Reply:

- We thank the reviewer for this comment, and have revised the whole manuscript to avoid language errors and to improve the English.

2. The research gap is not very clearly established. Are there any other studies that address the same aspect for India? What is the novelty of your work? Clearly highlight the research gap and the motivation.

Reply:

- Thanks for raising the question. The research gap has now been more clearly described. The major contributions of the paper lie in two aspects as follows:
 - On lines 70-83: “The primary challenge in planning for power systems under high levels of renewables is to reconcile the conflict between the variability of renewable sources and the intrinsic inflexibility of thermal power systems. This has led to a loss of more than \$10 billion for China due to curtailments of wind and solar power. Important contributions have been conducted for national level pathway studies [R1-R6]. However, the flexibility issue has been largely simplified, resulting in over optimistic projections of carbon abatement costs and under investments in flexible power generating resources. Here we introduce an integrated renewable energy system planning model designed to co-optimize investments for generation, transmission and storage expansion with detailed treatment of system operations

considering not only requirements for balancing supply and demand, but also hourly ramping, reserves, minimal load, and timing involved in start-up and shut down of thermal units. To accelerate the optimization process, our previously developed Fast Unit Commitment model is applied to reduce the computational complexity for operational simulation.”

- On lines 332-340: “The striking conclusion is that investments in renewables could play an important role in reducing the overall costs for delivery of electricity in India’s future power system. In particular, assuming a commitment to 80% renewables using the standard model, the overall costs for delivery of power in 2040 were projected to be lower by as much as \$50 billion compared with expenses for the 0% renewable reference case, equivalent to a decline of 0.5 cent/kWh for retail electricity prices.”

3. Are there any government policies that affect the energy planning decision? These need to be considered in the analysis.

Reply:

- Thanks for this comment. We now note the following policy implications:
 - On lines 102-107: “We choose to focus here on wind and solar recognizing the emphasis that has been placed on these resources most recently by the Indian government. Current policy calls for 175 GW of renewable energy by 2022, 160 GW of which would be supplied in the form of either wind or solar. Reports indicate that this focus on wind and solar is likely to continue and indeed most likely to expand beyond this initial target date [3].”
 - Also, there are several political strategies of Renewable Portfolio Standards (RPS), renewable certificate, investment credit, and other incentives for promoting renewables. Our analysis helps to establish improved targets for RPS and renewable certificates for different regions.
 - As for the continued government support to develop coal-fired power plants, “the locked-in capacity will affect the optimal generation mix as well as the system economics in the future” on lines 48-50.

4. I feel that in the background section preceded the model formulation, the flow of the paper would be clearer. In addition, the model formulation section (which seems to represent the methodology in the main body of the text) needs to be divided into sub-sections for clarity.

Reply:

- Thanks a lot for the great suggestions. We have reorganized the Model Formulation section entirely and removed unnecessary sections or moved them to the methodology (specifically Projected costs for each technology and Options for storage).

5. Some of the information, especially the statistics, provided under the model formulation section would be better suited in the background. In addition, some of the results are also included under the model formulation.

Reply:

- Thanks a lot for the great suggestions. We have reorganized the Model Formulation to

better fit with the rest of the main text.

6. The economic implications need to be further highlighted in the discussion. What does this mean for the energy costs incurred by the consumers at present?

Reply:

- Thanks for the suggestion. Transitioning away from fossil sources using wind and solar power would reduce the overall cost for electricity. In particular, assuming a commitment to 80% renewables using the standard model, the overall costs for delivery of power in 2040 were projected to be lower by as much as \$50 billion compared with expenses for the 0% renewable reference case. These changes could potentially reduce the energy costs incurred by consumers. As indicated in Table R1, incorporating higher penetration of renewables would lower the levelized cost of electricity by 0.8-0.5 cents/kWh depending on the renewable penetration level. We have added implications in the discussion section of the updated manuscript, on lines 329-338.

Table R1 levelized cost as a function of varying levels of investments in renewables

0%	20%	40%	60%	80%
0.0611\$/kWh	0.0532\$/kWh	0.0497\$/kWh	0.0518\$/kWh	0.0566\$/kWh

Page 2:

Ln 30-41: Perhaps this information could be presented in graphical form for further clarity.

Reply:

- Thanks for this comment. We now present this information in Supplementary Figure 1.

Page 3:

Ln 64-65: The sentence is not clear. Please revise.

Reply:

- Thanks for raising the question. We have elaborated the sentence for the modeling formulation, and expanded the description into a paragraph on lines 70-83.

Page 10

Ln 233-237: What does these results signify?

Reply:

- Thanks a lot for the comments on results related to storage and transmission expansion. Notable investments in transmission are sited between the North, East, South and West regions. Stronger transmission connection between these four regions could mitigate the variability of wind and solar power at larger geographical scope, significantly reducing the requirements for flexibility resources such as storage. We have incorporated the above analysis in lines 188-189 and 194-196.

Again, we thank the reviewer for the considerate comments. We hope our revisions address all the concerns raised.

References

- R1 Shukla, P. R. et al. Pathways to deep decarbonization in India, SDSN - IDDRI. (2015).
- R2 Gadre, R. & Anandarajah, G. Assessing the evolution of India's power sector to 2050 under different CO2 emissions rights allocation schemes. *Energy Sustain Dev*, 50, 126–138 (2019).
- R3 Tokimatsu, K., Yasuoka, R., & Nishio, M. Global zero emissions scenarios: The role of biomass energy with carbon capture and storage by forested land use. *App. Energy*, 185, 1899–1906 (2016).
- R4 Grubler, A., et al. A low energy demand scenario for meeting the 1.5C target and sustainable development goals without negative emissions technologies. *Nature Energy*, 3, 515–527 (2018).
- R5 Anandarajah, G. & Gambhir, A. India's CO2 emission pathways to 2050: what role can renewables play? *Appl. Energy*, 131, 79–86 (2014).
- R6 Mittal, S., Dai, H., Fujimori, S., & Masui, T. Bridging greenhouse gas emissions and renewable energy deployment target: comparative assessment of China and India. *Appl. Energy*, 166, 301–313 (2016).
- R7 NITI Aayog. India's Energy and Emissions Outlook: Results from India Energy Model. <https://niti.gov.in/sites/default/files/2019-07/India%E2%80%99s-Energy-and-Emissions-Outlook.pdf> (2017).
- R8 News report available at: <https://www.pv-magazine.com/2019/01/08/india-will-tender-500-gw-renewable-capacity-by-2028/>

REVIEWERS' COMMENTS:

Reviewer #2 (Remarks to the Author):

I feel that the research gap and the need for the study should be further established. Further, more highlight should be given to the policy implications of the study.

However, the other comments have been addressed satisfactorily.

Manuscript Number: NCOMMS-20-02366-T

Title: India's potential for integrating solar and on- and offshore wind power into its energy system

Response to Reviews

We would like to thank the editor and reviewers for the helpful comments and for the thorough review of our paper.

We thank the reviewers and the editor for the careful review of our paper. The detailed suggestions are important for improving its quality. We have addressed the research gap question suggested by Reviewer #2. Detailed responses to the reviewers are given below. All of the changes in the manuscript are indicated in red in the Additional Material file.

Response to Anonymous Reviewers

Reviewer: 2

Reviewer comments to the author:

I feel that the research gap and the need for the study should be further established. Further, more highlight should be given to the policy implications of the study.

However, the other comments have been addressed satisfactorily.

We thank the reviewer for the helpful comments throughout the review process. We have further addressed the research gap and policy implications in several sections, specifically:

- “There is a clear need for an integrated view of the potential for a low-carbon future in India. This paper represents for the first time an integrated view of all components of India’s electricity system involving wind, solar, hydro, coal, gas, storage and interregional transmission to meet power demand on hourly basis.” Lines 58-61.
- “Significant computational challenge is incurred when modeling the full set of flexibility requirements.” Lines 87-88.
- “The findings summarized here provide clear implications for energy and environmental policies for India’s electricity sector in planning for a low-carbon 2040.” Lines 345-347.